# A Deep Learning Trained Clear-Sky Mask Algorithm for VIIRS Radiometric Bias Assessment

**Xingming Liang [1,2,\*], Quanhua Liu [1], Banghua Yan [1] and Ninghai Sun [1,3]**

[1] Center for Satellite Applications and Research, National Environmental Satellite (STAR),
Data, and Information Service (NESDIS), National Oceanic and Atmospheric Administration (NOAA),
College Park, MD 20740, USA; quanhua.liu@noaa.gov (Q.L.); Banghua.Yan@noaa.gov (B.Y.);
Ninghai.Sun@noaa.gov (N.S.)

[2] Earth System Science Interdisciplinary Center, University of Maryland, College Park, MD 20740, USA

[3] Global Science and Technology, Inc. (GST), Greenbelt, MD 20770, USA

\* Correspondence: xingming.liang@noaa.gov

**Abstract:** Clear-sky mask (CSM) is a crucial influence on the calculating accuracy of the sensor radiometric biases for spectral bands of visible, infrared, and microwave regions. In this study, a fully connected deep neural network (FCDN) was proposed to generate CSM for the Visible Infrared Imaging Radiometer Suite (VIIRS) onboard Suomi National Polar-Orbiting Partnership (S-NPP) and NOAA-20 satellites. The model, well-trained by S-NPP data, was used to generate both S-NPP and NOAA-20 CSMs for the independent data, and the results were validated against the biases between the sensor observations and Community Radiative Transfer Model (CRTM) calculations (O-M). The preliminary result shows that the FCDN-CSM model works well for identifying clear-sky pixels. Both O-M mean biases and standard deviations were comparable with the Advance Clear-Sky Processor over Ocean (ACSPO) and were significantly better than a prototype cloud mask (PCM) and the case without a clear-sky check. In addition, by replacing CRTM brightness temperatures (BTs) with the atmosphere air temperature and water vapor contents as input features, the FCDN-CSM exhibits its potential to generate fast and accurate VIIRS CSM onboard follow-up Joint Polar Satellite System (JPSS) satellites for sensor calibration and validation before the physics-based CSM is available.

**Keywords:** clear-sky mask; fully connected deep neural network; VIIRS; bias assessment

## 1. Introduction

Sensor radiometric bias and stability are key to evaluating sensor calibration performance and cross-sensor consistency [1–6]. They also help to identify the root causes of Environment Data Record (EDR) or Level 2 product issues, such as sea surface temperature and cloud mask [1–3,7]. The bias characteristic is even used for radiative transfer model validation, such as the Community Radiative Transfer Model (CRTM) [2,8]. The sensor radiometric bias is commonly calculated by the difference between similar sensor observations using a simultaneous nadir overpass approach (SNO) [9,10], sensor observations minus radiative transfer model simulations [2,3], or sensor observations against the hyperspectral instrument measurements convoluted by a spectral response function [11]. Clear-sky mask (CSM) is critically important for reconciling these differences and improving the radiometric biases, with spectral regions covering visible to infrared, and even for cloud-insensitive microwave regions. Developed at NOAA STAR, the Integrated Calibration and Validation (CAL/VAL) System Long-Term Monitoring (ICVS, https://www.star.nesdis.noaa.gov/icvs/) and the Monitoring of Clear-sky Radiance over Ocean for Sea Surface Temperature (SST) (MICROS, https://www.star.nesdis.noaa.gov/sod/sst/micros/) are two typical examples used to monitor sensor observations minus CRTM simulation

biases (O-M), with clear-sky conditions for different sensors onboard NOAA related satellite series. Both webs show several tenths of Kelvin in the O-M mean biases and ~0.6 K in standard deviations for most of the surface emission bands after the clear-sky check, which is much more accurate than the all-sky condition.

Clear-sky mask products have been developed with different sensor measurements at NOAA and NASA, with examples including the NOAA Clouds from Advanced Very High Resolution Radiometer (AVHRR) Extended (CLAVR-x), the Moderate Resolution Imaging Spectroradiometer (MODIS) cloud mask, the Visible Infrared Imaging Radiometer Suite (VIIRS) cloud mask (VCM), and the Advanced Clear-Sky Processor over Ocean (ACSPO). The CSM products developed by the EDR teams are customarily based on the empirical threshold methods calculated from the radiative transfer theory and cloud physical characteristics. For instance, the ACSPO used eight empirical threshold tests to screen out most of the cloud pixels in the global ocean domain for SST retrieval [12] and make the O-M distribution more Gaussian and bring the value closer to 0. The well-established ACSPO CSM has been comprehensively used in SST and other remote sensing communities for various meteorology satellite sensors [2–4], particularly the new-generation NOAA satellite sensors, such as the VIIRS onboard Joint Polar Satellite System (JPSS) and Advanced Baseline Imager (ABI) onboard the Geostationary Operational Environmental Satellite-R Series (GOES-R).

With cutting-edge artificial intelligence (AI) evolving rapidly, deep learning (DL), one of the most popular AI methods, has made a remarkable difference in many science and engineering fields. Its application in remote sensing [13,14] and numerical weather prediction [15–18] is also being explored. Deep learning is constructed using artificial neural networks (ANNs), including more than one hidden layer, with a so-called "deep" neural network distinguished from a "shallow" neural network. The deep learning is efficient and straightforward and it uses massive remote-sensing data-trained coefficients to generate a product, in contrast to traditional physics-based methods. Different DL methods have been proposed based on different applications of learning [19,20]. The main methods include fully connected deep neural networks (FCDNs) and convolution neural networks (CNNs) [21]. Applying the deep learning method for CSM has been explored in the recent decade [22–24]. Most of these researches used CNN to identify cloud or clear-sky features from the satellite images in a regional area, and the model accuracy and efficiency were not validated globally. In this study, the FCDN method was employed to train and generate CSM in the global ocean domain for VIIRSs onboard S-NPP and NOAA-20, with the ACSPO CSM data used as references. The MICROS O-M tool was used to validate the FCDN-CSM performance through comparison with ACSPO, a prototype cloud mask, and all-sky condition. The objective was to develop a fast and straightforward CSM for VIIRS CAL/VAL and address the advantage of the DL method used for CSM classification. Section 2 summarized the basic concept of FCDN and its implementation in CSM retrieval. Section 3 analyzed the FCDN-CSM results. We first demonstrated the use of the S-NPP data to train the FCDN-CSM, then evaluated the FCDN-CSM results using O-M biases for both S-NPP and NOAA-20. In the next stage, we evaluated the long-term performance of the FCDN-CSM model by using a nine-day training dataset. Finally, we explored the potential features selection for a fast FCDN-CSM model. Section 4 discussed two specific advantages of the FCDN-CSM. Section 5 presents the conclusion.

## 2. FCDN-CSM Network Architecture

In this section, we give an overview of the basic concept of FCDN and present the input and output data. We then demonstrate the implementation of FCDN-CSM training and testing. For more details about FCDN, the reader should refer to the literature cited herein.

### 2.1. Basic Concept of FCDN

The FCDN is an ANN, one of the popular models of artificial intelligence used to efficiently solve problems of function fitting, classification, clustering, and pattern recognition. The basic concept of

the ANN is that the model mathematically simulates the work of the biological neural network in a straightforward way [25].

Like a biological neural network, the ANN is composed of a number of individual artificial neurons. Each artificial neuron is an individual computational unit, and it receives several input signals $X = [x1, x2, \ldots xn]$, called "features". Each input feature is arbitrarily given a weight $W = [w1, w2, \ldots wn]$, and each neuron is given a bias, $B$, to calculate one output value using a simple dot product and a nonlinear activation function, as follows:

$$Y = a(h) = a(W * X + B). \tag{1}$$

Here, $a(h)$ is activation function. It could be a sigmoid, rectified linear unit (ReLU), or Tanh function, depending on the kind of application used.

The individual neurons may simultaneously receive the input $X$ and generate outputs based on their specific weights and biases. These parallel neurons form a hidden layer (HL1). The generated output dataset from HL1 ($Y^{(1)}$) can be treated as a new input set to create HL2 and product $Y^{(2)}$. By repeating the above procedure, we can create as many hidden layers as possible, and for each hidden layer, we can add any number of neurons. The final layer outputs expected $\hat{Y}$, known as "output layer". The input data (referred to as input layer), hidden layers, and output layer form the architecture of the ANN (Figure 1). The neurons in each layer are fully connected to the neurons of the next layer. If the number of hidden layers is greater than one, this is known as an FCDN [26]. This is the most popular model of the ANN family adopted in this study, and it is also known as a "feedforward neural network" (FNN). The greater the number of layers and neurons in the FCDN, the greater the complexity and capability of the model, but the more computer time and storage are needed. Thus, it is crucial, when designing the FCDN architecture, to decide carefully the number of hidden layers and the number of neurons for each layer.

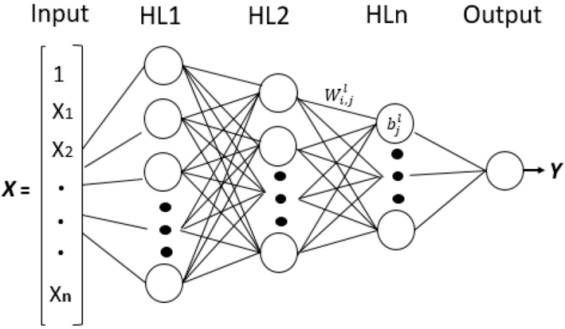

**Figure 1.** A scheme of fully connected "Deep" neural network.

After deciding the input data and ANN architecture, the model training is conducted to optimize the values for all the weights and biases in iterative processing. Many data samples, including known $X$ and "true" $Y$, must be collected and fed into the model during the training. The number of samples should usually be one order larger than the number of free parameters to ensure that those parameters are adequate to be optimized. Each iteration includes two-step procedures: forward propagation and backward propagation. In forward propagation, each neuron output value is calculated using input data from the input layer or output from the last HL, together with assumed weights and biases from the left side of the ANN architecture (Figure 1) to the right. After going through all layers, the final value of the output layer, the expected $\hat{Y}$, is produced. The expected $\hat{Y}$ is then compared with the "true" $Y$ to calculate the error by a cost function, of which the selection is also dependent on the application. As CSM is a classification problem in this application, we used the cross-entropy loss as the cost function (Equation (2)):

$$J(\hat{y},\ y) = \frac{1}{N}\sum_{n=1}^{N}(-\text{y}\log\hat{y} + (1-y)\log(1-\hat{y})) \tag{2}$$

Here, $N$ is the number of input samples. The backward propagation optimizes weights and biases by minimizing the cost function (Equation (2)). Each weight and bias is adjusted repeatedly using *Stochastic* Gradient Descent (SGD) or a similar method (Equations (3) and (4)) from the right side of Figure 1 to the left, which is the reverse of the order in forward propagation.

$$W_{m+1}^{l} = W_{m}^{l} - \beta\frac{\partial J}{\partial W_{m}^{l}} \tag{3}$$

$$b_{m+1}^{l} = b_{m}^{l} - \beta\frac{\partial J}{\partial b_{m}^{l}} \tag{4}$$

Here, $l$ is the layer number, and $m$ is the iteration number. $\beta$ stands for the learning rate, and this is one of the key hyper-parameters of the FCDN algorithm. The selection of $\beta$ significantly affects the efficiency of the model convergence. The adjusted weights and biases are used in the next iteration, together with the newly selected samples, as input to calculate the next step of the cost function and to optimize weights and biases again. This processing repeats until the cost has been minimized to an acceptable error and the optimized weights and biases are obtained. During the model training, the trained weights and biases are tested using a test dataset. The objective of the test dataset is to track the optimal status of the weights and biases and ensure cost convergence. The comparison of the errors in the test dataset and the training dataset provides an excellent indicator of whether the model is optimized, underfit, or overfit. We will further analyze the training and test errors in the next section.

## 2.2. Data Input and Output

The ACSPO CSM for VIIRS is based on eight threshold tests to identify CS pixels using five major data sources: (i) three brightness temperatures (BTs) of thermal emission bands (TEB)-M12, M15, and M16; (ii) corresponding CRTM BTs; (iii) sensor geometrical parameters (solar zenith angle and satellite zenith angle); (iv) regression and reference SSTs; and v) SST spatial variance. The regression SST is obtained from the nonlinear SST algorithm [12]. The Canadian Meteorological Centre (CMC) L4 0.1-degree analysis SST (https://podaac.jpl.nasa.gov/dataset/CMC0.1deg-CMC-L4-GLOB-v3.0) was adopted into ACSPO as a reference SST. The CRTM BTs were calculated using the CRTM version 2.3.0, together with the quarter degree Global Forecasting System (GFS) atmospheric profiles and CMC SST as inputs, which is used indirectly in ACSPO for CSM retrieval [12]. The SST spatial variances were defined by the standard deviations calculated from a three-by-three window cell around the analyzed pixel. All data are well integrated into the ACSPO analysis product to retrieve three types of clear-sky masks: clear-sky, probable clear-sky (PCS), and cloud. The clear-sky is defined for SST retrieval to release as much global coverage for the SST community as possible, including of in-land water and many coastal areas, which does not help to calculate sensor radiometric biases accurately. Nevertheless, ACSPO also saves an individual filter for the brightness temperature test, which is a more clear-sky conservative for sensor O-M biases [27]. Thus, in this research, we separated the clear-sky type into two categories. The first was clear-sky for BT(CS_BT), including the clear-sky pixels that passed the BT test. The remaining clear-sky pixels were referred to as "clear-sky for SST" (CS_SST). For sensor CAL/VAL, the CS_BT is more relevant. Overall, for initial testing, all five data source and a total of 11 features were selected as the input into the FCDN-CSM model, together with four types of CSM product, which were used as labels and lists as follows:

Input features: [M12_BT, M15_BT, M16_BT, M12_CRTM_BT, M15_CRTM_BT, M16_CRTM_BT, Regression SST, Reference SST, Solar_Zenith_Angle, Satellite_Zenith_Angle, Spatial_Variance].

Output CSM types: [CS_BT, CS_SST, PCS, CLOUD].

To minimize the solar contamination and diurnal cycle effect [28], the nighttime CSM was tested first. Comparison with the historical sensor—the Advanced Very High Resolution Radiometer

(AVHRR)—revealed that the VIIRS had a new band (M14 (8.5 μm)), and the analysis of this band was added to the newest ACSPO version. We used ACSPO V2.40 data for this research, which does not include M14 analysis, but ACSPO V2.40 has been well-validated, and the CSM produced is also comparable with the later ACSPO version. Version 2.40 was thus adequate for this research, which focuses only on FCDN-CSM performance. The analysis of M14 will be added in future research on improving CSM, and the results will be published elsewhere. In addition, only ocean pixels were available for this research, as ACSPO was developed for SST retrieval.

### 2.3. FCDN-CSM Architecture

After selecting the input data, the most critical step was to create the FCDN architecture, including deciding the number of hidden layers and the number of neurons in each layer, the activation function, and the other hyper-parameters of FCDN, such as learning rate. There is no mathematical or physical rule to determine these parameters, other than early ANN references and repeated experiments. As the number of input features was small (11) and the number of output clear-sky types was just four, two hidden layers were deemed sufficient to approximate most of the function [29]. Since the CSM is a nonlinear classification problem, we used ReLU as the activation function for HL1 to ensure nonlinear processing and used the cross-entropy loss as a cost function in the output layer to ensure optimal classification. As is well known, a high learning rate may speed up model convergence in the early iteration period, but this often results in model divergence, since the updated step for gradient descent is large in the later iteration period. On the other hand, a slower learning rate will result in the convergence taking too long due to the small step. Thus, we used an exponentially decaying learning rate to avoid this potential issue (https://www.tensorflow.org/api_docs/python/tf/train/exponential_decay). The primary metrics to decide the number of neurons in the two hidden layers concern the avoidance of overfitting and underfitting. The number of neurons in each layer can be further optimized to reduce the model complexity, compute time, and memory. Based on several experiments, we ultimately decided on 40 neurons for HL1 and 90 for HL2, and we set 0.01 as the initial learning rate.

The optimization of the weights and biases is an iterative procedure. In each iteration, the weights and biases are updated with their partial derivative to the cost function as Equations (3) and (4) during the backward propagation processing, and the cost function is calculated using the updated weights and biases in the forward propagation, with new samples fed into the input layer. The method of feeding the data into every iteration is a vital hyper-parameter. Commonly, the number of the selected samples used for training (namely, one epoch) is vast, and it is impossible to feed all samples into each iteration, as this would demand huge computer time and memory. However, feeding individual samples into the model would create more noises for each iteration, making the model convergence take too long and meaning it would be near impossible to reach an optimal value. Thus, a small batch of samples was selected for every iteration, namely the "mini-batch." The mini-batch may not always minimize the cost function, since it is selected differently for each iteration, but a well-selected mini-batch will ultimately make cost function converge with a global minimum, although it will oscillate during the iteration period. We selected 256 data samples as a mini-batch to feed into each iteration. The acceptable convergence condition was determined as the point at which iteration processing could be stopped to obtain an acceptable result. We also saved the best result during the iteration processing to ensure global optimization.

Model testing is a necessary procedure during the training period to identify any problems (such as underfitting and overfitting) and modify the model accordingly. For this work, we developed a special validation algorithm to check how well the model was working arbitrarily over time during the training period by automatically transiting the free parameters from the training part to validation. When a core deep network is well trained, it can be employed to predict the CSM of independent data, as will be demonstrated in the following section.

## 3. Results

### 3.1. FCDN-CSM Training and Testing Using S-NPP Data

More than 100 million S-NPP VIIRS pixels for the period of 1–10 March 2019, were chosen as the training dataset, with the snow, ice, and all invalid values removed. There were 20,000 data samples with the same period as the training set, with different pixels selected as a test dataset. During the model training period, we randomly chose 2% of the training data as a validation dataset. The preprocessing of input data is necessary to guarantee the accuracy and efficiency of the model calculation. First, to avoid regional biases, the input data were randomly shuffled before being separated into the training and test datasets. Second, for the training dataset, four CSM types were selected, with a similar percentage to ensure every type had sufficient data samples fed into the model. This selection criterion was also used for the O-M validation in the later subsections. Finally, all train data were normalized to the extent of a mean bias of 0 and a standard deviation of 1 to ensure the input dataset was consistent and to speed up the convergence of the cost function [30,31].

The model was adequately trained, and the weights and biases were well-optimized after 200,000 iterations. Figure 2 shows the convergence of the cost function (blue curve) and the prediction accuracy (red curve). Each point in the blue curve is the value of the cost function (cross-entropy loss) per 1000 iterations, and each point in the red curve is the accuracy, also defined as recall rate for all types between the predicted CSM and "true", recorded every 10 s. The cost function from 1.4 was rapidly reduced to 0.3 at the beginning of 5000 iterations and then gradually reduced to ~0.17 during 195,000 iterations. Similarly, the accuracy rose quickly from 78% to 94%, then gradually approached 96%. Table 1 shows the recalls and precisions for the test dataset for individual CSM type and the accuracy of all four types. The calculation of the recall and precision is from the confusion matrix [32]. The accuracy for the test dataset was slightly less than—though still close to—the training set. The recall rates of the CS_BT, PCS, and CLOUD types, and the precisions of the CS_BT and CLOUD types were all above 92%, indicating that the model worked well and had no significant underfitting and overfitting, although the precisions for the PCS and CS_SST were low, maybe due to the unbalance of the four CSM types [32]. On the other hand, the recall rate for CS_SST was ~7% lower than that of the other three types. As described in the previous section, the CS_SST is defined such that ACSPO releases more SST pixels for better global coverage to meet user requirements. More features are likely to be needed to improve the matchup accuracy for the CS_SST type. Nevertheless, the CS_BT is more conservative for the clear-sky condition, which is the best fit for sensor calibration and validation.

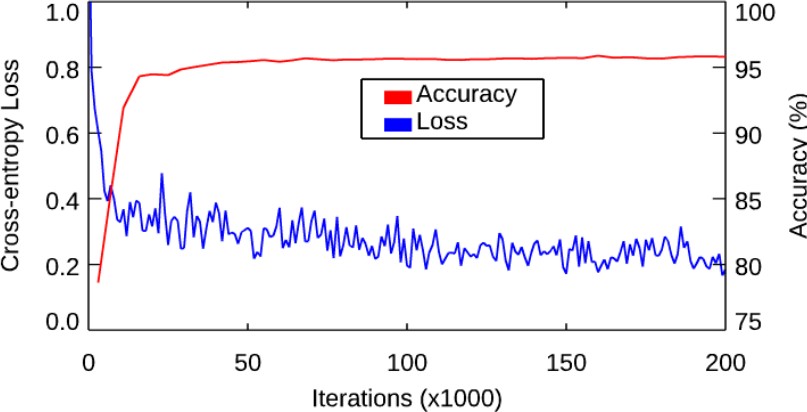

**Figure 2.** Changes of loss and accuracy during fully connected deep neural network-clear-sky mask (FCDN-CSM) training. The selected Suomi National Polar-Orbiting Partnership (S-NPP) Visible Infrared Imaging Radiometer Suite (VIIRS) sensor data records (SDR) data for the days of 1–10 March 2019, were fed into the model. The left Y-axis corresponds to the change of loss, and the right Y-axis corresponds to the change of accuracy.

The well-trained weights and biases were then applied to generate S-NPP CSM for 11 March 2019. Figure 3 shows the global distribution of the four CSM types between ACSPO and FCDN. As in Table 1, the global distribution for CS_BT, PCS, and CLOUD matched well with ACSPO, but the CS_SST pixels were a little more mismatched than others, such as the west and middle of the Pacific Ocean, the north and middle of the Arabian Sea, and the African coastal area.

**Table 1.** The recalls and precisions of the test dataset of four CSM data types.

| | SNPP | | | |
|---|---|---|---|---|
| | **ACSPO N** | **ML N** | **Recall (%)** | **Precision (%)** |
| CS_BT | 3348 | 3139 | 93.76 | 92.70 |
| PCS | 703 | 674 | 95.87 | 86.19 |
| CLOUD | 15,575 | 150,50 | 96.63 | 98.52 |
| CS_SST | 374 | 330 | 88.24 | 80.29 |
| ALL | 20,000 | 19,193 | 95.97 | 96.67 |

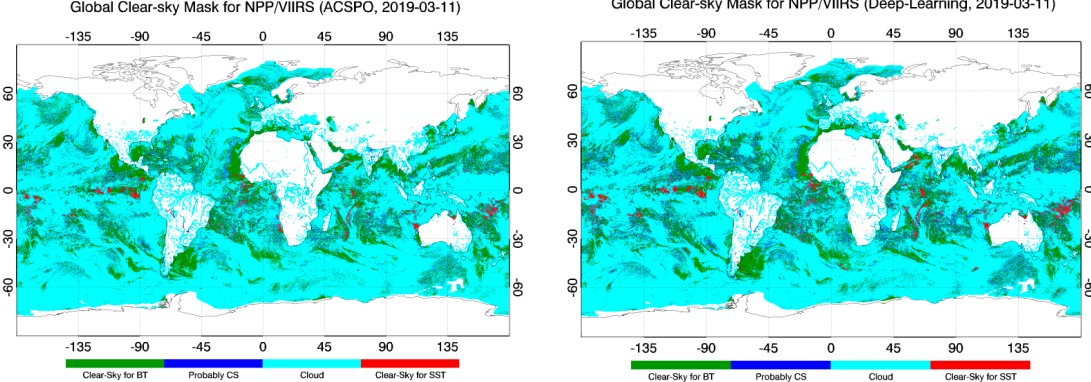

**Figure 3.** Global distribution of the four CSM types from Advance Clear-Sky Processor over Ocean (ACSPO) (**left**) and FCDN-CSM (**right**) for S-NPP.

*3.2. FCDN-CSM Validation Using O-M Biases*

The sensor observation minus CRTM simulation (O-M) biases were applied to identify issues of sensor calibration, to validate CRTM, and to check cross-platform consistency and CSM during the past decade [1–4]. The O-M biases were first monitored at the NOAA monitoring of clear-sky radiances over the ocean for SST (MICROS, https://www.star.nesdis.noaa.gov/sod/sst/micros/) in 2008. The O-M validation was then extended to Integrated Calibration/Validation System Long-Term Monitoring (ICVS, https://www.star.nesdis.noaa.gov/icvs/) for sensor CAL/VAL, particularly VIIRS.

In this work, the O-M tool was used to validate the FCDN-CSM model. Additionally, we developed a prototype cloud detection (PCM) algorithm, which was similar to the early cloud detection product—Clouds from the AVHRR-Phase I (CLAVR-1) [29]. Four tests were used in the PCM: the Thermal Gross Contrast Test (TGCT), the Uniform Low Stratus Test (ULST), the "Four Minus Five" Test (FMFT), and the Nighttime Cirrus Test (CIRT). The detailed calculation information is well documented in Stowe et al. [33]. The objective of PCM is to verify the FCDN-CSM model, in addition to ACSPO.

Figure 4 shows the histograms of the O-M mean biases in M12, with the CS pixels identified by the PCM, ACSPO, and FCDN-CSM, and without CS checks (all-sky condition). "*Y*-axis" denotes the ratio of the number of samples in a bin and the total number of samples. Table 2 shows the corresponding global O-M mean biases and standard deviations (STDs) for five TEB M-bands and the number of clear-sky pixels (NCSP). As described in the previous section, the CS_BT type in ACSPO and FCDN-CSM were used to check the clear-sky pixels to provide more accurate O-M biases for the VIIRS calibration. In Figure 4, the histogram for the all-sky case has a long tail on the left, which is attributed to cloud contamination. This resulted in significant negative O-M biases, with large

STDs (approximately −13 ± 16 K, Table 2) for all bands, as CRTM simulation is under the clear-sky condition, and the model BTs would be overestimated when the simulated pixel was under cloud conditions. The PCM screened out a large proportion of the cloud pixels and reduced the O-M statistics to ~−1.8 K ± 2.3 K while retaining a "negative" tail. However, both ACSPO and FCDN-CSM showed very similar mean biases, STDs, and number of clear-sky pixels. Both removed most cloud pixels and made the histogram significantly narrower and closer to the Gauss distribution. The global mean biases and STDs were much smaller than those of the PCM for all five bands. Notably, the mean and STD in M12 were −0.15 ± 0.35 K for both ACSPO and FCDN-CSM with comparison to approximately −1.4 ± 2.4 K for PCM. Note that the physical mechanisms causing cold O-M biases for all bands, discussed in Liang et al. [1,2], are due to missing aerosol and using CMC SST without diurnal cycle correction in the model, leaving "M" slightly overestimated. The possible residual cloud in the ACSPO and FCDN also decreases the "O" term, amplifying the negative shift in the O-M bias.

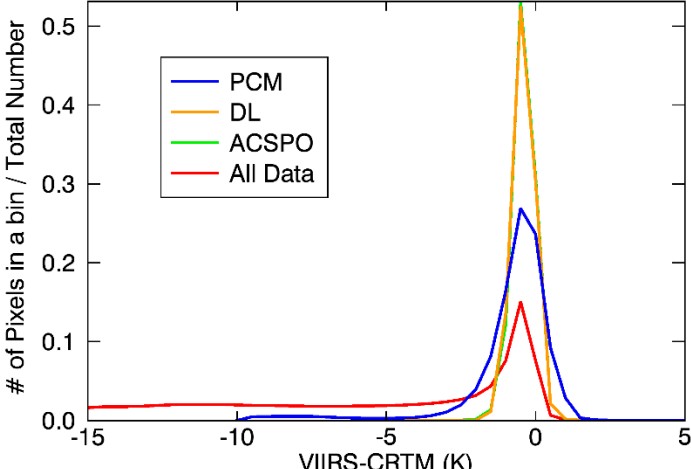

**Figure 4.** Histograms of VIIRS M12 O-M biases for the cases of PCM, ACSPO, FCDN-CSM, and all data on 11 March 2019. The Y values stand for the ratios between the number of samples in a bin with total number.

**Table 2.** Global O-M statistics for S-NPP on 11 March 2019. (μ stands for O-M bias and σ is standard deviation).

|  | ACSPO | | FCDN-CSM | | PCM | | ALL Data | |
|---|---|---|---|---|---|---|---|---|
|  | μ | σ | μ | σ | μ | σ | μ | σ |
| M12 | −0.1349 | 0.3479 | −0.1453 | 0.3486 | −1.3868 | 2.3835 | −11.6833 | 14.0627 |
| M13 | −0.5873 | 0.3554 | −0.5858 | 0.3581 | −1.8384 | 2.1222 | −11.4019 | 13.7826 |
| M14 | −0.7165 | 0.4499 | −0.7027 | 0.4421 | −1.6244 | 1.9611 | −12.6647 | 14.7198 |
| M15 | −0.5912 | 0.4896 | −0.5689 | 0.4778 | −1.4705 | 2.1007 | −13.4887 | 15.8929 |
| M16 | −0.7312 | 0.5756 | −0.7000 | 0.5532 | −1.4670 | 1.9744 | −13.5286 | 15.7936 |
| NCSP × 10$^4$ | 2111 | | 2123 | | 3862 | | 12,570 | |

### 3.3. NOAA-20 CSM Retrieval Using FCDN-CSM

Since the instrument structure and radiometric performance between S-NPP VIIRS and NOAA-20 VIIRS are similar [34], the S-NPP data-trained FCDN-CSM may be applied to generate NOAA-20 CSM. To validate this, NOAA-20 VIIRS SDR data for 11 March 2019, were collected, with sensor geophysical and surface parameters. The collected data were used to generate NOAA-20 CSM, using S-NPP-trained FCDN-CSM. We then compared the O-M biases for NOAA-20, using the CSM generated by the FCDN-CSM and the ACSPO V2.60, a newer ACSPO version compatible with NOAA-20. Figure 5 shows the global distribution of the CSM types between ACSPO v2.60 and FCDN-CSM-generated NOAA-20 CSM. The O-M biases for M12, M14, M15, and M16 of NOAA-20 are listed in Table 3.

Note that the analysis for M13 is not available in the official ACSPO V2.60. As expected, the global distribution of generated NOAA-20 CSM was comparable with that of ACSPO V2.60. Similarly, for all four bands, the mean biases and STDs were comparable, thus verifying that the S-NPP-trained CSM can be used to generate accurate NOAA-20 CSM.

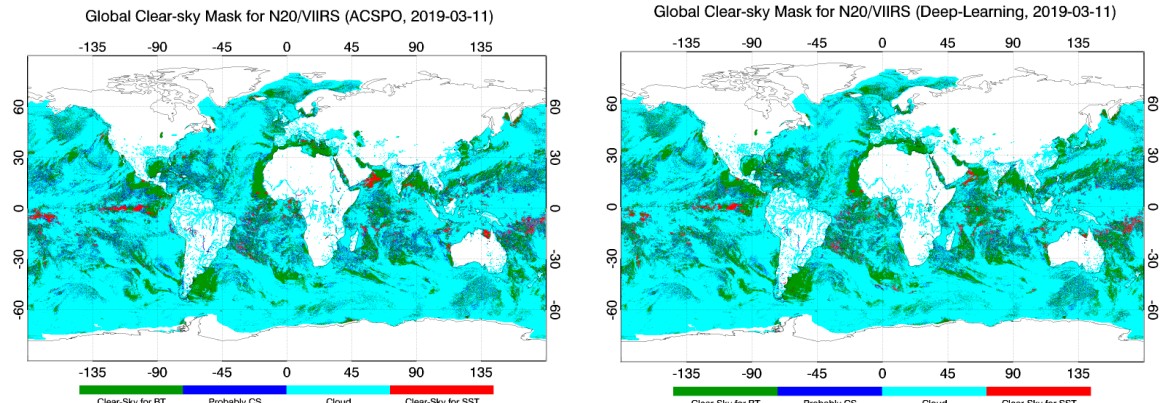

**Figure 5.** Global distribution of the four CSM types from ACSPO (**left**) and FCDN-CSM (**right**) for NOAA-20.

**Table 3.** Global O-M statistics for NOAA-20 on 11 March 2019. (μ stands for O-M bias and σ is standard deviation).

|  | ACSPO | | FCDN-CSM | |
| --- | --- | --- | --- | --- |
|  | μ | σ | μ | σ |
| M12 | −0.2412 | 0.3559 | −0.1990 | 0.3538 |
| M14 | −0.7378 | 0.4509 | −0.6120 | 0.4553 |
| M15 | −0.5676 | 0.4796 | −0.5366 | 0.4892 |
| M16 | −0.6936 | 0.5754 | −0.6773 | 0.5871 |
| NCSP × $10^4$ | 2075 | | 2017 | |

### 3.4. Stability of the FCDN-CSM

In this section, nine days (11–19 March 2019) of S-NPP CSMs were generated by the FCDN-CSM model to evaluate the model performance by checking the stability and accuracy of the O-M biases. Figure 6 shows the O-M error bars with STDs for the five TEB M-bands, using ACSPO CSM and FCDN-CSM to identify clear-sky pixels. The O-M mean, STDs, and NCSP were stable and comparable with ACSPO in all bands for all nine days. The differences of the mean and STDs were within 0.01 ± 0.02 K, and the NCSP differences were less than 0.5% for all bands and the whole period, indicating that the FCDN-CSM model is robust and stable and can work well for a short-term period. All nine-day statistics results are considerably consistent, implying the global CSMs are stable day-to-day. Further analysis is underway by selecting some discrete days of data in a year to evaluate the long-term performance of the model and check whether the FCDN-CSM is affected by the seasonal cycle. The result will be published elsewhere.

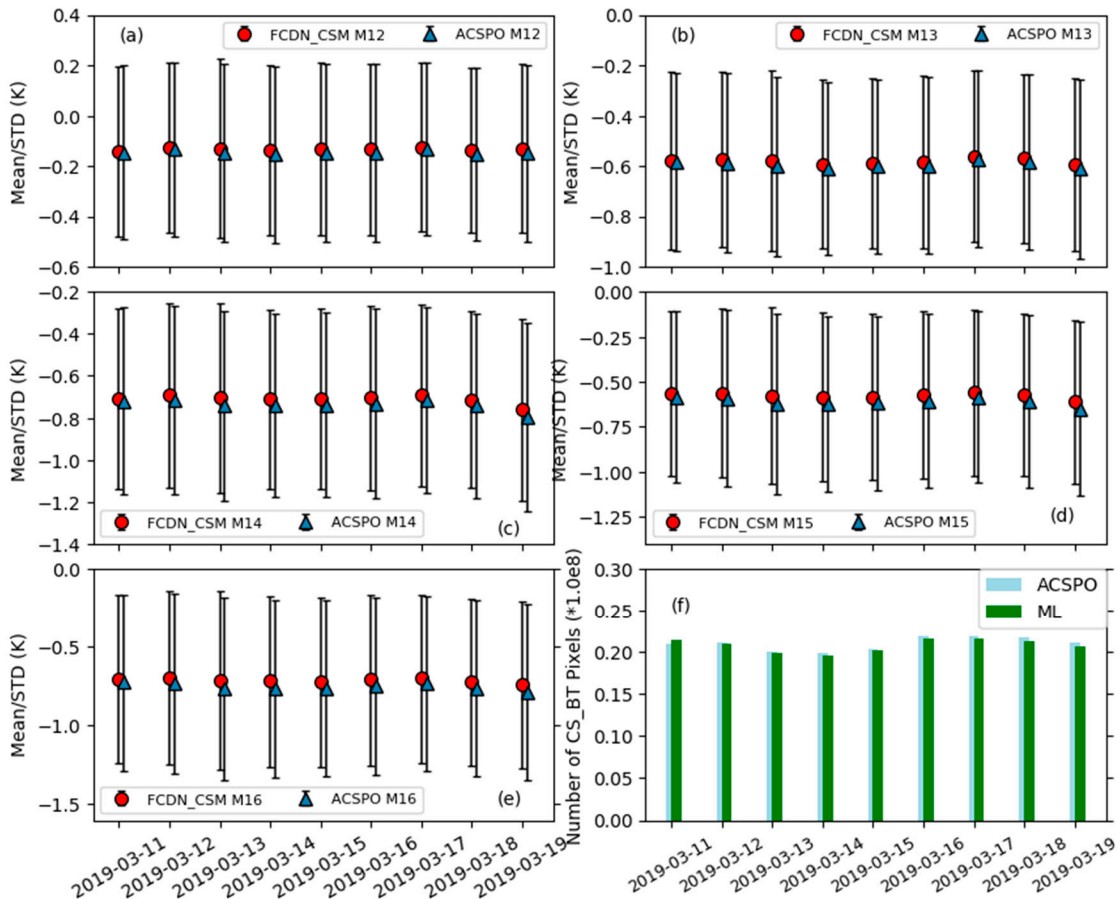

**Figure 6.** The M-O mean and STD error bars by using ACSPO CSM and FCDN_CSM for nine-day data from 11 to 19 March 2019 for (**a**) M12, (**b**) M13, (**c**) M14, (**d**) M15, and (**e**) M16, and corresponding NCSPs (**f**).

### 3.5. Selection of Important Features

As is well known, the use of three CRTM BTs (M12, M15, and M16) is computationally expensive for CSM retrieval, as the CRTM calculation is time-consuming. It is necessary to explore other features to replace CRTM BTs as model input before implementing FCDN-CSM for VIIRS CAL/VAL. Theoretically, introducing CRTM BTs is more likely to use atmosphere and surface states for CSM retrieval, since the CRTM simulation uses atmosphere profiles, surface temperature, and emissivity as input, particularly for atmosphere temperature and water vapor profiles, which are considered to be the direct causes to affect atmosphere clear-sky and cloud status. Two further tests were conducted to check this hypothesis. Both tests removed the three CRTM BTs from the input layer. One test added column water vapor contents (CWV), surface air temperature, and surface water vapor contents as three new features (referred to 11-updated-feature) to replace CRTM BTs. The other test did not add any other feature (8-feature). Figure 7 shows the convergences of the loss function for both cases and compares them to the original, which included the three CRTM BTs as the input features (11-feature). The cost functions finally converged at 0.28, 0.22, and 0.17 in the cases of the 8-feature, 11-updated-feature, and 11-feature, respectively. This suggests that the CWV and surface air states have similar capabilities to the three CRTM BTs used as FCDN-CSM input to identify the clear-sky mask. Table 4 summarizes the recalls and precisions with ACSPO CSM for the two cases. As expected, the recall and precision in the case of the eight features were worse for all four types compared to that of the 11-feature listed in Table 1. For instance, the CS_BT, most concerned in this paper, were reduced by ~13%. On the other hand, adding CWV, surface air temperature, and surface water vapor content increased the recall accuracy by ~8% compared to the 8-feature, and so did for the precision, which confirms that the atmosphere states

could potentially replace the CRTM calculation in the FCDN-CSM model. For CS_SST, the degradation of the recall rate for the 8-feature was more significant. As discussed in the previous section, the issue for CS_SST is out of the scope of this research, and we will analyze this issue in future work. Table 5 compares the O-M biases using the CSM generated by the two cases. Comparison with the 11-feature listed in Table 2 reveals that the O-M mean biases were comparable for all five bands in all three cases. However, the STDs for the case without CRTM BTs (8-feature) were 0.03, 0.02, 0.08, 0.10, and 0.11 K larger than that with CRTM BTs (11-feature) for each band, respectively, suggesting that the atmosphere states are critically important for CSM retrieval. However, the 11-updated-feature significantly reduced STDs, which were only up to 0.03 K degraded from the 11-feature for all five bands. This verifies that the CWV, surface air temperature, and water vapor contents could be used in FCDN-CSM, rather than the time-consuming CRTM BTs, and the contribution to CSM of surface emissivity is minimal. Using NOAA internal Linux server with 100 G memory and 2.2 G multi-core CPUs and without GPU support, the FCDN-CSM takes less than one minute to generate one day of CSM (about 0.6 billion pixels), including the calculation of CWV and other atmosphere parameters, whereas the ACSPO needs more than five hours to obtain the same CSM product. Overall, the 11-updated-feature is considered a fast and accurate CSM model for VIIRS CAL/VAL.

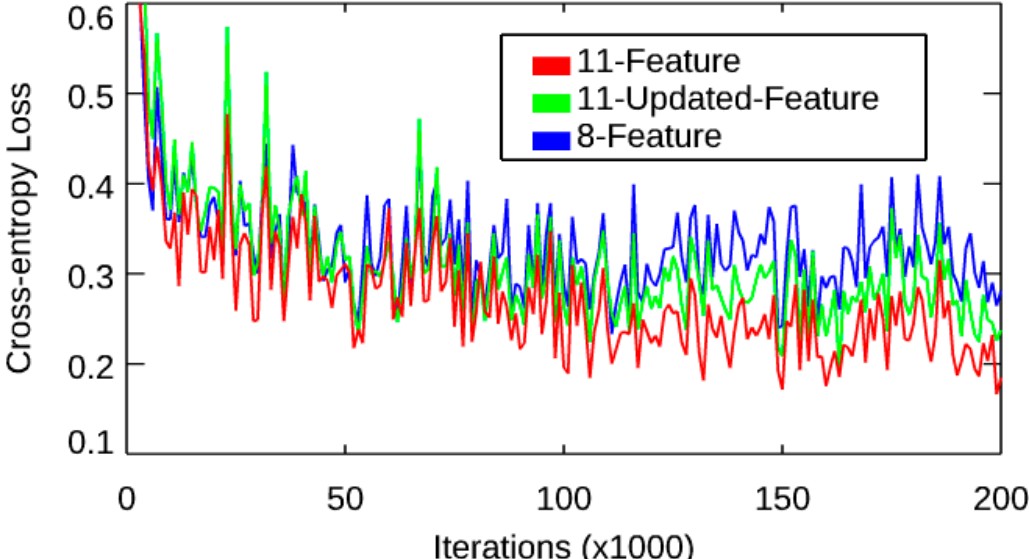

**Figure 7.** The changes of the cost for the cases of 11-feature (including 3 Community Radiative Transfer (CRTM) brightness temperatures (BTs)), 8-feature (excluding 3 CRTM BTs), and 11-updated-feature (excluding 3 CRTM BTs but adding CWV, surface air temperature, and water vapor contents as new features).

**Table 4.** Recalls and precisions of the test dataset for the cases without CRTM BT input (8-feature) and without CRTM BT but with CWV as input (11-updated-Feature).

|  | **8-Feature** | | | **11-Updated-Feature** | | |
|---|---|---|---|---|---|---|
|  | **N** | **Recall (%)** | **Precision (%)** | **N** | **Recall (%)** | **Precision (%)** |
| CS_BT | 2775 | 82.89 | 78.26 | 3037 | 90.71 | 91.28 |
| PCS | 677 | 96.30 | 63.99 | 676 | 96.16 | 85.35 |
| CLOUD | 14,918 | 95.78 | 95.23 | 14,957 | 96.03 | 98.14 |
| CS_SST | 247 | 66.04 | 58.67 | 291 | 77.81 | 81.74 |
| Accuracy | 18,617 | 93.09 | 89.98 | 18,961 | 94.81 | 96.18 |

**Table 5.** The O-M biases for 8-feature and 11-updated-Feature on 11 March 2019. (μ stands for O-M bias and σ is standard deviation).

|  | 8-Feature | | 11-Updated-Feature | |
|---|---|---|---|---|
|  | μ | σ | μ | σ |
| M12 | −0.1541 | 0.3784 | −0.1393 | 0.3600 |
| M13 | −0.5605 | 0.3728 | −0.5829 | 0.3599 |
| M14 | −0.6961 | 0.5243 | −0.6950 | 0.4620 |
| M15 | −0.5486 | 0.5849 | −0.5603 | 0.4969 |
| M16 | −0.6841 | 0.6888 | −0.6898 | 0.5852 |
| NCSP $\times 10^4$ | 1964 | | 2083 | |

## 4. Discussion

In the last section, we compared the FCDN-CSM results with other CSMs as well as analyzing the O-M biases. The validation showed a generally good agreement between the FCDN-CSM and the ACSPO CSM. In this section, we discussed several specific features and science concerns in the FCDN-CSM.

The first feature is the migration advantage of the FCDN-CSM. Commonly, different CSM product is developed using different sensor SDR or level 1 data, and provided by the EDR team. This sensor-independent CSM is not available during the sensor data records (SDR) calibration and validation period, immediately after the sensor launch. Although a proxy CSM with a similar sensor can be used initially to validate O-M biases, the O-M statistics are generally not as good as expected. For instance, in the NOAA ICVS web page (https://www.star.nesdis.noaa.gov/icvs/status_N20_VIIRS.php), the monitoring of the O-M biases for NOAA20 VIIRS uses S-NPP cloud mask as a proxy, resulting the standard deviations are ~8% larger than S-NPP. However, Section 3.4 demonstrated that using S-NPP data trained FCDN-CSM can apply for NOAA-20, and the O-M mean biases and STDs with FCDN-generated NOAA-20 CSM were still comparable with ACSPO. This experiment indicated that the FCDN-CSM model might be a better proxy CSM for the follow-up JPSS satellites to evaluate O-M biases for sensor CAL/VAL before the EDR CSM product is released.

The objective of this research is to develop a fast and accurate clear-sky mask for VIIRS radiometric bias assessment. Specifically, assuring the accuracy and stability of the O-M bias by accurately identifying clear-sky pixels is our scope in this paper. Thus, the accuracy of the CS_BT and the model efficiency are the most concerns, instead of the accuracies of all CSM types. Table 1 showed more than 92% recall and precision in CS_BT, which justified that the FCDN-CSM can identify CS pixels accurately. Although there were still 6–8% inconsistent for the CS_BT between FCDN-CSM and ACSPO, the fact that the O-M mean biases and STDs were not getting worse compared to ACSPO in Table 2 indicated that the misidentification of CS_BT maybe not always wrong in the FCDN-CSM. The potential residual cloud in the ACSPO CSM is also a contributor [1]. Furthermore, the nine-day O-M statistics analysis in Section 3.4 verified that the FCDN-CSM is robust and accurate compared to ACSPO.

Finally, as discussed in Section 3.5, using the CWV, surface air temperature, and water vapor contents in FCDN-CSM instead of time-consuming CRTM BTs did not significantly reduce the recalls and precisions for four CSM types, but made the FCDN-CSM processing much more efficient than ACSPO.

## 5. Conclusions and Future Work

A FCDN-CSM model was introduced to explore the improvement of the O-M biases for the sensor CAL/VAL, particularly for VIIRS SDR radiometric biases accuracy and efficiency. The model was constructed with an 11-feature input layer, two hidden layers with 40 by 90 neurons, and an output layer with four clear sky and cloud types. The model selected more than one hundred million nighttime samples from ten days of ACSPO S-NPP VIIRS data as input, separated into training and testing datasets. The model was trained and validated iteratively to optimize weights and biases.

The well-trained model was then used to generate both S-NPP and NOAA-20 CSMs, and the results were compared with ACSPO, PCM, and the cloud model, using the O-M biases validation.

The O-M biases for FCDN-CSM were comparable with ACSPO for all five VIIRS TEB bands. Both FCDN-CSM and ACSPO CSM were more accurate than PCM and the case without CS check, indicating that FCDN-CSM works well for generating both S-NPP and NOAA-20 CSMs. The nine-day time series of O-M biases using FCDN-CSM was comparable to ACSPO, confirming that the FCDN-CSM is robust and stable for a short-term period. The CWV and surface air temperature and water vapor contents were selected to replace the time-consuming CRTM-BTs as input, verifying that the model could be used as a fast and accurate CSM for follow-up VIIRS CAL/VAL. Future work should validate the FCDN-CSM algorithm for land conditions using the VIIRS Cloud Mask product [35]. It is expected that the well-validated algorithm could advance the accuracy of VIIRS O-M biases for NOAA-20 and future JPSS VIIRS.

**Author Contributions:** X.L. developed methods, created the evaluation dataset, analyzed results and wrote the manuscript. Q.L. and B.Y. supervised research. N.S. provided technical help for the project. All authors have read and agreed to the published version of the manuscript.

**Acknowledgments:** This work is funded by the JPSS Programs and CICS projects. The ACSPO and CRTM are provided by SST and CRTM teams of NOAA STAR. Thanks go to Alexander Ignatov of STAR SST Team, Changyong Cao, Slawomir Blonski and all of the VIIRS SDR Team, and Sid Boukabara of the STAR AI Group for advice and help. The views, opinions, and findings contained in this report are those of the authors and should not be construed as an official NOAA or U.S. Government position, policy, or decision.

**Conflicts of Interest:** The authors declare no conflict of interest.

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
