# Peer review of "A Deep Learning Trained Clear-Sky Mask Algorithm for VIIRS Radiometric Bias Assessment"

_remotesensing, doi:10.3390/rs12010078_

Round 1
Reviewer 1 Report
General Comments:
This paper introduces an approach for generating clear-sky mask for VIIRS instruments using deep neural network. Data from S-NPP VIIRS was used to train the model. The model was then used to generate CSMs for S-NPP and NOAA-20. The results were validated by comparing the (O-M) bias calculated between the different models and CTRM calculation.
Overall, the presentation of the idea in the paper is clear and well organized. While the methodology in the paper is new and deserve publication, the results in the paper has no noticeable differences over the current one (ACSPO). More discussion may be needed to discuss the advantages of the proposed model in comparison with ACSPO. There are series of minor issues that the author needs to address.
Minor Issues:
Is the use of AI to generete CSMs in this paper published the first time? If not, how does it compare with the published? Do you have any results that compare the results with VIIRS cloud mask product. Line 166-174 and line 175-183 is same. Line 395- “FCDN-CSM is robust and stable” : You have just validated the CSMs for 9 days, I think you need to test it for longer time to make this “robust” claim.
Author Response
Dear Reviewer1,
Thank you for your detailed review and creative comments. Please find our point-to-point responses in the attached file.
Sincerely,
Xingming Liang

Reviewer 2 Report
To mask out cloudy pixels from assessing the sensor radiometric bias, this paper introduced a machine learning (ML) based clear-sky mask method. A fully connected two-hidden-layer neural network was built and trained by using the S-NPP VIIRS pixels and ACSP CSM obtained from March 1st - 10th in 2019. It was then followed by comparing the retrieved CSM to the other CSMs as well as analyzing the Observation-Model biases. The validation shows a generally good agreement between the ML-based CSM and the ACSP CSM. To further investigate the potential of the trained model, Liang et al. also applied it to NOAA-20 VIIRS and tested the feasibility of using meteorological variables to replace CRTM BTs. Both experiments suggested the proposed 11-feature ML CSM model is a fast and accurate model for VIIRS CAL/VAL.
The manuscript is overall well written and self-explained. Some of the results are interesting and important for future cloud mask studies using DL approaches. However, there are parts that were not addressed properly/enough in the current manuscript. I therefore suggest publication of the manuscript after addressing the following questions.
General questions
Regarding the ML cloud/clear masks, my biggest concern is the stability of the algorithm and the fact that the model cannot perform better than the given training dataset. We know that cloud mask (CM) is basically a purpose-driven product, which varies between the cloud-conservative CM and the clear-conservative CM. According to lines 153-155, the referenced ASCPO CSM seems like a cloud-conservative CM. Given the accuracies listed in Table 1, is the ML CSM more towards to clear-conservative or cloud-conservative, or varies case-by-case? I’m also confused about both Tables 1 and 4, as well as the definition of matchup rate (Line 244). What’s the mathematical expression of the matchup rate and its relationship to the recall and precision that calculated from the confusion matrix? Or just replacing Tables 1 and 4 by the confusion matrix?
Moreover, I’m curious about the advantage of the ML-based CSM comparing to the ASCPO CSM, as the training dataset is ASCPO and he ML CSM cannot perform better than it. The authors did address one migration advantage by apply S-NPP trained model to NOAA-20 and compared results with the ACSPO V2.6. However, the authors had not justified their arguments (Lines 324-325) enough. One way to justify it is probably to apply S-NPP's CSM algorithm to NOAA-20 for the same data. The other potential advantage as mentioned in Line 397 (used as a fast ...) was also not justified in the manuscript. Given a comparison of the runtime between ML and ASCPO algorithms could be helpful.
Specific comments
Line 57, “one of the most popular AI methods”
Line 92, “W•X” rather than “W.X”
Line 102, the reference to the statement “... hidden layers is greater than one, this is known as an FCDN” is missi
Line 109, Figure 1, remove red underlines in the figure
Line 125, is it Eq. 2 or Eq. 3?
Line 165, PCS is not specified, does it mean “probable clear-sky"?
Line 175-183, duplicate paragraph
Line 190, “clear-sky types was just four”
Line 227, “for the period of March 1-10, 2019”
Line 228, does “example” also mean sample (same as Line 215)? As mentioned in Line 227, there were more than 100 million valid values. Why not used more samples to train the model. The statement is not clear here. Are these 20,000 “examples” the same as the “examples” in Table 1? Moreover, where is the validation data? The authors need to clearly specify the separation of training, validation, and test datasets here, otherwise, the following results are not convincing.
Line 232, rate -> percentage?
Line 264, Figure 3, labels are not clear
Line 267, “... during the past decade”?
Line 280, any reason not using “Y-axis”?
Line 303, Table 2, specify the definition of µ and sigma (same for Tables 3 and 5)
Line 328, Section 3.4, the section is not convincing. Basically 9-day period is not long enough to justify the arguments made in this Section. Authors may consider either extend the study period to include one seasonal cycle (different solar zenith angles) or remove this part.
Line 338, Figure 6, the figure is too low quality to visual.
Author Response
Dear Reviewer,
Thank you for your detailed review and creative comments. Please find our point-to-point responses in the following attached file.
Sincerely,
Xingming Liang

Round 2
Reviewer 2 Report
The authors have well addressed most of my previous concerns in a proper way. Therefore I suggest publishing as it is.